# STRUCTURED NEURAL SUMMARIZATION

**Patrick Fernandes, Miltiadis Allamanis & Marc Brockschmidt**
Microsoft Research
Cambridge, United Kingdom
{t-pafern,miallama,mabrocks}@microsoft.com

## ABSTRACT

Summarization of long sequences into a concise statement is a core problem in natural language processing, requiring non-trivial understanding of the input. Based on the promising results of graph neural networks on highly structured data, we develop a framework to extend existing sequence encoders with a graph component that can reason about long-distance relationships in weakly structured data such as text. In an extensive evaluation, we show that the resulting hybrid sequence-graph models outperform both pure sequence models as well as pure graph models on a range of summarization tasks.

## 1 INTRODUCTION

Summarization, the task of condensing a large and complex input into a smaller representation that retains the core semantics of the input, is a classical task for natural language processing systems. Automatic summarization requires a machine learning component to identify important entities and relationships between them, while ignoring redundancies and common concepts.

Current approaches to summarization are based on the sequence-to-sequence paradigm over the words of some text, with a sequence encoder — typically a recurrent neural network, but sometimes a 1D-CNN (Narayan et al., 2018) or using self-attention (McCann et al., 2018) — processing the input and a sequence decoder generating the output. Recent successful implementations of this paradigm have substantially improved performance by focusing on the decoder, extending it with an attention mechanism over the input sequence and copying facilities (See et al., 2017; McCann et al., 2018). However, while standard encoders (*e.g.* bidirectional LSTMs) theoretically have the ability to handle arbitrary long-distance relationships, in practice they often fail to correctly handle long texts and are easily distracted by simple noise (Jia & Liang, 2017).

In this work, we focus on an improvement of sequence encoders that is compatible with a wide range of decoder choices. To mitigate the long-distance relationship problem, we draw inspiration from recent work on highly-structured objects (Li et al., 2015; Kipf & Welling, 2017; Gilmer et al., 2017; Allamanis et al., 2018; Cvitkovic et al., 2018). In this line of work, highly-structured data such as entity relationships, molecules and programs is modelled using graphs. Graph neural networks are then successfully applied to directly learn from these graph representations. Here, we propose to extend this idea to weakly-structured data such as natural language. Using existing tools, we can annotate (accepting some noise) such data with additional relationships (*e.g.* co-references) to obtain a graph. However, the sequential aspect of the input data is still rich in meaning, and thus we propose a hybrid model in which a standard sequence encoder generates rich input for a graph neural network. In our experiments, the resulting combination outperforms baselines that use pure sequence or pure graph-based representations.

Briefly, the contributions of our work are: 1. A framework that extends standard sequence encoder models with a graph component that leverages additional structure in sequence data. 2. Application of this extension to a range of existing sequence models and an extensive evaluation on three summarization tasks from the literature. 3. We release all used code and data at https://github.com/CoderPat/structured-neural-summarization.

```
public void Add(string name, object value = null, DbType? dbType = null,
            ParameterDirection? direction = null, int? size = null,
            byte? precision = null, byte? scale =null) {
  parameters[Clean(name)] = new ParamInfo{
    Name = name, Value = value,
    ParameterDirection = direction ?? ParameterDirection.Input,
    DbType = dbType, Size = size, Precision = precision, Scale = scale
  };}
```

| | |
|---|---|
| **Ground truth**: | add a parameter to this dynamic parameter list |
| **BiLSTM → LSTM**: | adds a new parameter to the specified parameter |
| **BiLSTM+GNN → LSTM**: | creates a new instance of the dynamic type specified |
| **BiLSTM+GNN → LSTM+Pointer**: | add a parameter to a list of parameters |

Figure 1: An example from the dataset for the METHODDOC source code summarization task along with the outputs of a baseline and our models. In the METHODNAMING dataset, this method appears as a sample requiring to predict the name `Add` as a subtoken sequence of length 1.

## 2  STRUCTURED SUMMARIZATION TASKS

In this work, we consider three summarization tasks with different properties. All tasks follow the common pattern of translating a long (structured) sequence into a shorter sequence while trying to preserve as much meaning as possible. The first two tasks are related to the summarization of source code (Figure 1), which is highly structured and thus can profit most from models that can take advantage of this structure; the final task is a classical natural language task illustrating that hybrid sequence-graph models are applicable for less structured inputs as well.

**METHODNAMING**   The aim of this task is to infer the name of a function (or method in object-oriented languages, such as Java, Python and C#) given its source code (Allamanis et al., 2016). Although method names are a single token, they are usually composed of one or more subtokens (split using `snake_case` or `camelCase`) and thus, the method naming task can be cast as predicting a sequence of subtokens. Consequently, method names represent an "extreme" summary of the functionality of a given function (on average, the names in the Java dataset have only 2.9 subtokens). Notably, the vocabulary of tokens used in names is very large (due to abbreviations and domain-specific jargon), but this is mitigated by the fact that 33% of subtokens in names can be copied directly from subtokens in the method's source code. Finally, source code is highly structured input data with known semantics, which can be exploited to support name prediction.

**METHODDOC**   Similar to the first task, the aim of this task is to predict a succinct description of the functionality of a method given its source code (Barone & Sennrich, 2017). Such descriptions usually appear as documentation of methods (*e.g.* "docstrings" in Python or "JavaDocs" in Java). While the task shares many characteristics with the METHODNAMING task, the target sequence is substantially longer (on average 19.1 tokens in our C# dataset) and only 19.4% of tokens in the documentation can be copied from the code. While method documentation is nearer to standard natural language than method names, it mixes project-specific jargon, code segments and often describes non-functional aspects of the code, such as performance characteristics and design considerations.

**NLSUMMARIZATION**   Finally, we consider the classic summarization of natural language as widely studied in NLP research. Specifically, we are interested in abstractive summarization, where given some text input (*e.g.* a news article) a machine learning model produces a novel natural language summary. Traditionally, NLP summarization methods treat text as a sequence of sentences and each one of them as a sequence of words (tokens). The input data has less explicitly defined structure than our first two tasks. However, we recast the task as a structured summarization problem by considering additional linguistic structure, including named entities and entity coreferences as inferred by existing NLP tools.

## 3 MODEL

As discussed above, standard neural approaches to summarization follow the sequence-to-sequence framework. In this setting, most decoders only require a representation $\boldsymbol{h}$ of the complete input sequence (*e.g.* the final state of an RNN) and per-token representations $\boldsymbol{h}_{t_i}$ for each input token $t_i$. These token representations are then used as the "memories" of an attention mechanism (Bahdanau et al., 2014; Luong et al., 2015) or a pointer network (Vinyals et al., 2015a).

In this work, we propose an extension of sequence encoders that allows us to leverage known (or inferred) relationships among elements in the input data. To achieve that, we combine sequence encoders with graph neural networks (GNNs) (Li et al., 2015; Gilmer et al., 2017; Kipf & Welling, 2017). For this, we first use a standard sequential encoder (*e.g.* bidirectional RNNs) to obtain a per-token representation $\boldsymbol{h}_{t_i}$, which we then feed into a GNN as the initial node representations. The resulting per-node (*i.e.* per-token) representations $\boldsymbol{h}'_{t_i}$ can then be used by an unmodified decoder. Experimentally, we found this to surpass models that use either only the sequential structure or only the graph structure (see Sect. 4). We now discuss the different parts of our model in detail.

**Gated Graph Neural Networks** To process graphs, we follow Li et al. (2015) and briefly summarize the core concepts of GGNNs here. A graph $\mathcal{G} = (\mathcal{V}, \mathcal{E}, \boldsymbol{X})$ is composed of a set of nodes $\mathcal{V}$, node features $\boldsymbol{X}$, and a list of directed edge sets $\boldsymbol{\mathcal{E}} = (\mathcal{E}_1, \ldots, \mathcal{E}_K)$ where $K$ is the number of edge types. Each $v \in \mathcal{V}$ is associated with a real-valued vector $\boldsymbol{x}_v$ representing the features of the node (*e.g.*, the embedding of a string label of that node), which is used for the initial state $\boldsymbol{h}_v^{(0)}$ of a node.

Information is propagated through the graph using neural message passing (Gilmer et al., 2017). For this, every node $v$ sends messages to its neighbors by transforming its current representation $\boldsymbol{h}_v^{(i)}$ using an edge-type dependent function $f_k$. Here, $f_k$ can be an arbitrary function; we use a simple linear layer. By computing all messages at the same time, all states can be updated simultaneously. In particular, a new state for a node $v$ is computed by aggregating all incoming messages as $\boldsymbol{m}_v^{(i)} = g(\{f_k(\boldsymbol{h}_u^{(i)}) \mid \text{there is an edge of type } k \text{ from } u \text{ to } v\})$. $g$ is an aggregation function; we use elementwise summation for $g$. Given the aggregated message $\boldsymbol{m}_v^{(i)}$ and the current state vector $\boldsymbol{h}_v^{(i)}$ of node $v$, we can compute the new state $\boldsymbol{h}_v^{(i+1)} = \text{GRU}(\boldsymbol{m}_v^{(i)}, \boldsymbol{h}_v^{(i)})$, where GRU is the recurrent cell function of a gated recurrent unit. These dynamics are rolled out for a fixed number of timesteps $T$, and the state vectors resulting from the final step are used as output node representations, *i.e.*, $\text{GNN}((\mathcal{V}, \boldsymbol{\mathcal{E}}, \boldsymbol{X})) = \{\boldsymbol{h}_v^{(T)}\}_{v \in \mathcal{V}}$.

**Sequence GNNs** We now explain our novel combination of GGNNs and standard sequence encoders. As input, we take a sequence $S = [s_1 \ldots s_N]$ and $K$ binary relationships $R_1 \ldots R_K \in S \times S$ between elements of the sequence. For example, $R_=$ could be the equality relationship $\{(s_i, s_j) \mid s_i = s_j\}$. The choice and construction of relationships is dataset-dependent, and will be discussed in detail in Sect. 4. Given any sequence encoder $\mathcal{SE}$ that maps $S$ to per-element representations $[\mathbf{e}_1 \ldots \mathbf{e}_N]$ and a sequence representation $\mathbf{e}$ (*e.g.* a bidirectional RNN), we can construct the sequence GNN $\mathcal{SE}_{GNN}$ by simply computing $[\mathbf{e}'_1 \ldots \mathbf{e}'_N] = \text{GNN}((S, [R_1 \ldots R_K], [\mathbf{e}_1 \ldots \mathbf{e}_N]))$. To obtain a graph-level representation, we use the weighted averaging mechanism from Gilmer et al. (2017). Concretely, for each node $v$ in the graph, we compute a weight $\sigma(w(\boldsymbol{h}_v^{(T)})) \in [0, 1]$ using a learnable function $w$ and the logistic sigmoid $\sigma$ and compute a graph-level representation as $\hat{\mathbf{e}} = \sum_{1 \leq i \leq N} \sigma(w(\mathbf{e}'_i)) \cdot \aleph(\mathbf{e}'_i)$, where $\aleph$ is another learnable projection function. We found that best results were achieved by computing the final $\mathbf{e}'$ as $W \cdot (\mathbf{e}\, \hat{\mathbf{e}})$ for some learnable matrix $W$.

This method can easily be extended to support additional nodes not present in the original sequence $S$ after running $\mathcal{SE}$ (*e.g.*, to accommodate meta-nodes representing sentences, or non-terminal nodes from a syntax tree). The initial node representation for these additional nodes can come from other sources, such as a simple embedding of their label.

**Implementation Details.** Processing large graphs of different shapes efficiently requires to overcome some engineering challenges. For example, the CNN/DM corpus has (on average) about 900 nodes per graph. To allow efficient computation, we use the trick of Allamanis et al. (2018) where all graphs in a minibatch are "flattened" into a single graph with multiple disconnected components. The varying graph sizes also represent a problem for the attention and copying mechanisms in the

decoder, as they require to compute a softmax over a variable-sized list of memories. To handle this efficiently without padding, we associate each node in the (flattened) "batch" graph with the index of the sample in the minibatch from which the node originated. Then, using TensorFlow's `unsorted_segment_*` operations, we can perform an efficient and numerically stable softmax over the variable number of representations of the nodes of each graph.

## 4 EVALUATION

### 4.1 QUANTITATIVE EVALUATION

We evaluate Sequence GNNs on our three tasks by comparing them to models that use only sequence or graph information, as well as by comparing them to task-specific baselines. We discuss the three tasks, their respective baselines and how we present the data to the models (including the relationships considered in the graph component) next before analyzing the results.

### 4.1.1 SETUP FOR METHODNAMING

**Datasets, Metrics, and Models.**    We consider two datasets for the METHODNAMING task. First, we consider the "Java (small)" dataset of Alon et al. (2018a), re-using the train-validation-test splits they have picked. We additionally generated a new dataset from 23 open-source C# projects mined from GitHub (see below for the reasons for this second dataset), removing any duplicates. More information about these datasets can be found in Appendix C. We follow earlier work on METHOD-NAMING (Allamanis et al., 2016; Alon et al., 2018a) and measure performance using the F1 score over the generated subtokens. However, since the task can be viewed as a form of (extreme) summarization, we also report ROUGE-2 and ROUGE-L scores (Lin, 2004), which we believe to be additional useful indicators for the quality of results. ROUGE-1 is omitted since it is equivalent to F1 score. We note that there is no widely accepted metric for this task and further work identifying the most appropriate metric is required.

We compare to the current state of the art (Alon et al., 2018a), as well as a sequence-to-sequence implementation from the OpenNMT project (Klein et al.). Concretely, we combine two encoders (a bidirectional LSTM encoder with 1 layer and 256 hidden units, and its sequence GNN extension with 128 hidden units unrolled over 8 timesteps) with two decoders (an LSTM decoder with 1 layer and 256 hidden units with attention over the input sequence, and an extension using a pointer network-style copying mechanism (Vinyals et al., 2015a)). Additionally, we consider self-attention as an alternative to RNN-based sequence encoding architectures. For this, we use the Transformer (Vaswani et al., 2017) implementation in OpenNMT (*i.e.*, using self-attention both for the decoder and the encoder) as a baseline and compare it to a version whose encoder is extended with a GNN component.

**Data Representation**    Following the work of Allamanis et al. (2016); Alon et al. (2018a), we break up all identifier tokens (*i.e.* variables, methods, classes, *etc.*) in the source code into subtokens by splitting them according to `camelCase` and `pascal_case` heuristics. This allows the models to extract information from the information-rich subtoken structure, and ensures that a copying mechanism in the decoder can directly copy relevant subtokens, something that we found to be very effective for this task. All models are provided with all (sub)tokens belonging to the source code of a method, including its declaration, with the actual method name replaced by a placeholder symbol.

To construct a graph from the (sub)tokens, we implement a simplified form of the work of Allamanis et al. (2018). First, we introduce additional nodes for each (full) identifier token, and connect the constituent subtokens appearing in the input sequence using a INTOKEN edge; we additionally connect these nodes using a NEXTTOKEN edge. We also add nodes for the parse tree and use edges to indicate that one node is a CHILD of another. Finally, we add LASTLEXICALUSE edges to connect identifiers to their most (lexically) recent use in the source code.

### 4.1.2 SETUP FOR METHODDOC

**Datasets, Metrics, and Models.**    We tried to evaluate on the Python dataset of Barone & Sennrich (2017) that contains pairs of method declarations and their documentation ("docstring"). However, following the work of Lopes et al. (2017), we found extensive duplication between different folds

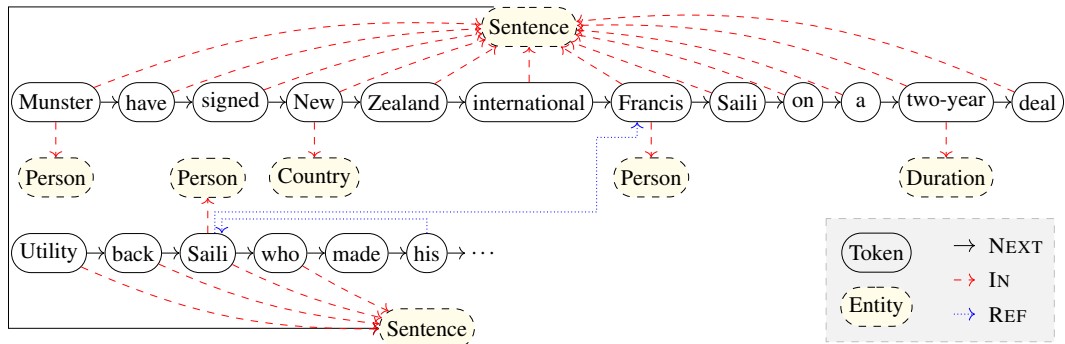

Figure 2: (Partial) graph of an example input from the CNN/DM corpus.

of the dataset and were only able to reach comparable results by substantially overfitting to the training data that overlapped with the test set. We have documented details in subsection C.3 and in Allamanis (2018), and decided to instead evaluate on our new dataset of 23 open-source C# projects from above, again removing duplicates and methods without documentation. Following Barone & Sennrich (2017), we measure the BLEU score for all models. However, we also report F1, ROUGE-2 and ROUGE-L scores, which should better reflect the summarization aspect of the task. We consider the same models as for the METHODNAMING task, using the same configuration, and use the same data representation.

### 4.1.3 SETUP FOR NLSUMMARIZATION

**Datasets, Metrics, and Models.** We use the CNN/DM dataset (Hermann et al., 2015) using the exact data and split provided by See et al. (2017). The data is constructed from CNN and Daily Mail news articles along with a few sentences that summarize each article. To measure performance, we use the standard ROUGE metrics. We compare our model with the near-to-state-of-the-art work of See et al. (2017), who use a sequence-to-sequence model with attention and copying as basis, but have additionally substantially improved the decoder component. As our contribution is entirely on the encoder side and our model uses a standard sequence decoder, we are *not* expecting to outperform more recent models that introduce substantial novelty in the structure or training objective of the decoder (Chen & Bansal, 2018; Narayan et al., 2018). Again, we evaluate our contribution using an OpenNMT-based encoder/decoder combination. Concretely, we use a bidirectional LSTM encoder with 1 layer and 256 hidden units, and its sequence GNN extension with 128 hidden units unrolled over 8 timesteps. As decoder, we use an LSTM with 1 layer and 256 hidden units with attention over the input sequence, and an extension using a pointer network-style copying mechanism.

**Data Representation** We use Stanford CoreNLP (Manning et al., 2014) (version 3.9.1) to tokenize the text and provide the resulting tokens to the encoder. For the graph construction (Figure 2), we extract the named entities and run coreference resolution using CoreNLP. We connect tokens using a NEXT edge and introduce additional super-nodes for each sentence, connecting each token to the corresponding sentence-node using a IN edge. We also connect subsequent sentence-nodes using a NEXT edge. Then, for each multi-token named entity we create a new node, labeling it with the type of the entity and connecting it with all tokens referring to that entity using an IN edge. Finally, coreferences of entities are connected with a special REF edge. Figure 2 shows a partial graph for an article in the CNN/DM dataset. The goal of this graph construction process is to explicitly annotate important relationships that can be useful for summarization. We note that (a) in early efforts we experimented with adding dependency parse edges, but found that they do not provide significant benefits and (b) that since we retrieve the annotations from CoreNLP, they can contain errors and thus, the performance of the our method is influenced by the accuracy of the upstream annotators of named entities and coreferences.

Table 1: Evaluation results for all models and tasks.

| METHODNAMING | F1 | ROUGE-2 | ROUGE-L | |
|---|---|---|---|---|
| Java | | | | |
| Alon et al. (2018a) | 43.0 | – | – | |
| SELFATT → SELFATT | 24.9 | 8.3 | 27.4 | |
| SELFATT+GNN → SELFATT | 44.5 | 20.9 | 43.4 | |
| BILSTM → LSTM | 35.8 | 17.9 | 39.7 | |
| BILSTM+GNN → LSTM | 44.7 | 21.1 | 43.1 | |
| BILSTM → LSTM+POINTER | 42.5 | 22.4 | 45.6 | |
| GNN → LSTM+POINTER | 50.5 | 24.8 | 48.9 | |
| BILSTM+GNN → LSTM+POINTER | **51.4** | **25.0** | **50.0** | |
| C# | | | | |
| SELFATT → SELFATT | 41.3 | 25.2 | 43.2 | |
| SELFATT+GNN → SELFATT | 62.1 | 31.0 | 61.1 | |
| BILSTM → LSTM | 48.8 | **32.8** | 51.8 | |
| BILSTM+GNN → LSTM | 62.6 | 31.0 | 61.3 | |
| BILSTM → LSTM+POINTER | 57.2 | 29.7 | 60.4 | |
| GNN → LSTM+POINTER | 63.0 | 31.5 | 61.3 | |
| BILSTM+GNN → LSTM+POINTER | **63.4** | 31.9 | **62.4** | |
| METHODDOC | F1 | ROUGE-2 | ROUGE-L | BLEU |
| C# | | | | |
| SELFATT → SELFATT | 40.0 | 27.8 | **41.1** | 13.9 |
| SELFATT+GNN → SELFATT | 37.6 | 25.6 | 37.9 | 21.4 |
| BILSTM → LSTM | 35.2 | 15.3 | 30.8 | 10.0 |
| BILSTM+GNN → LSTM | 41.1 | 28.9 | 41.0 | **22.5** |
| BILSTM → LSTM+POINTER | 35.2 | 20.8 | 36.7 | 14.7 |
| GNN → LSTM+POINTER | 38.9 | 25.6 | 37.1 | 17.7 |
| BILSTM+GNN → LSTM+POINTER (average pooling) | 43.2 | **29.0** | 41.0 | 21.3 |
| BILSTM+GNN → LSTM+POINTER | **45.4** | 28.3 | **41.1** | 22.2 |
| NLSUMMARIZATION | ROUGE-1 | ROUGE-2 | ROUGE-L | |
| CNN/DM | | | | |
| BILSTM → LSTM | 33.6 | 11.4 | 27.9 | |
| BILSTM+GNN → LSTM | 33.0 | 13.3 | 28.3 | |
| See et al. (2017) (+ Pointer) | 36.4 | 15.7 | 33.4 | |
| BILSTM → LSTM+POINTER | 35.9 | 13.9 | 30.3 | |
| BILSTM+GNN → LSTM+POINTER | 38.1 | 16.1 | 33.2 | |
| See et al. (2017) (+ Pointer + Coverage) | **39.5** | **17.3** | **36.4** | |

### 4.1.4 RESULTS & ANALYSIS

We show all results in Tab. 1. Results for models from the literature are taken from the respective papers and repeated here. Across all tasks, the results show the advantage of our hybrid sequence GNN encoders over pure sequence encoders.

On METHODNAMING, we can see that all GNN-augmented models are able to outperform the current specialized state of the art, requiring only simple graph structure that can easily be obtained using existing parsers for a programming language. The results in performance between the different encoder and decoder configurations nicely show that their effects are largely orthogonal.

On METHODDOC, the unmodified SELFATT → SELFATT model already performs quite well, and the augmentation with graph data only improves the BLEU score and worsens the results on ROUGE. Inspection of the results shows that this is due to the length of predictions. Whereas the ground truth data has on average 19 tokens in each result, SELFATT → SELFATT predicts on average 11 tokens, and SELFATT+GNN → SELFATT 16 tokens. Additionally, we experimented with an ablation in which a model is *only* using graph information, *e.g.*, a setting comparable to a simplification of the architecture of Allamanis et al. (2018). For this, we configured the GNN to use 128-dimensional representations and unrolled it for 10 timesteps, keeping the decoder configuration as for the other models. The results indicate that this configuration performs less well than a pure sequenced model. We speculate that this is mainly due to the fact that 10 timesteps are insufficient to propagate infor-

Table 2: Ablations on CNN/DM Corpus

| NLSUMMARIZATION (CNN/DM) | ROUGE-1 | ROUGE-2 | ROUGE-L |
|---|---|---|---|
| See et al. (2017) (base) | 31.3 | 11.8 | 28.8 |
| See et al. (2017) (+ Pointer) | 36.4 | 15.7 | 33.4 |
| See et al. (2017) (+ Pointer + Coverage) | **39.5** | **17.3** | **36.4** |
| BILSTM → LSTM | 33.6 | 11.4 | 27.9 |
| BILSTM → LSTM+POINTER | 35.9 | 13.9 | 30.3 |
| BILSTM → LSTM+POINTER (+ coref/entity annotations) | 36.2 | 14.2 | 30.5 |
| BILSTM+GNN → LSTM | 33.0 | 13.3 | 28.3 |
| BILSTM+GNN → LSTM+POINTER (only sentence nodes) | 36.0 | 15.2 | 29.6 |
| BILSTM+GNN → LSTM+POINTER (sentence nodes + eq edges) | 36.1 | 15.4 | 30.3 |
| BILSTM+GNN → LSTM+POINTER | 38.1 | 16.1 | 33.2 |

```
public static bool TryFormat(float value, Span<byte> destination,
      out int bytesWritten, StandardFormat format=default) {
   return TryFormatFloatingPoint<float>(value, destination,
                                   out bytesWritten, format); }
```

| | |
|---|---|
| **Ground truth** | formats a single as a utf8 string |
| **BILSTM → LSTM** | formats a number of bytes in a utf8 string |
| **BILSTM+GNN → LSTM** | formats a timespan as a utf8 string |
| **BILSTM+GNN → LSTM+POINTER** | formats a float as a utf8 string |

Figure 3: An example from the dataset for the METHODDOC source code summarization task along with the outputs of a baseline and our models.

mation across the whole graph, especially in combination with summation as aggregation function for messages in graph information propagation.

Finally, on NLSUMMARIZATION, our experiments show that the same model suitable for tasks on highly structured code is competitive with specialized models for natural language tasks. While there is still a gap to the best configuration of See et al. (2017) (and an even larger one to more recent work in the area), we believe that this is entirely due to our simplistic decoder and training objective, and that our contribution can be combined with these advances.

In Table 2 we show some ablations for NLSUMMARIZATION. As we use the same hyperparameters across all datasets and tasks, we additionally perform an experiment with the model of See et al. (2017) (as implemented in OpenNMT) but using our settings. The results achieved by these baselines trend to be a bit worse than the results reported in the original paper, which we believe is due to a lack of hyperparameter optimization for this task. We then evaluated how much the additional linguistic structure provided by CoreNLP helps. First, we add the coreference and entity annotations to the baseline BILSTM → LSTM + POINTER model (by extending the embedding of tokens with an embedding of the entity information, and inserting fresh "¡REF1¿", ... tokens at the sources/targets of co-references) and observe only minimal improvements. This suggests that our graph-based encoder is better-suited to exploit additional structured information compared to a biLSTM encoder. We then drop all linguistic structure information from our model, keeping only the sentence edges/nodes. This still improves on the baseline BILSTM → LSTM + POINTER model (in the ROUGE-2 score), suggesting that the GNN still yields improvements in the absence of linguistic structure. Finally, we add long-range dependency edges by connecting tokens with equivalent string representations of their stems and observe further minor improvements, indicating that even using only purely syntactical information, without a semantic parse, can already provide gains.

## 4.2 QUALITATIVE EVALUATION

We look at a few sample suggestions in our dataset across the tasks. Here we highlight some observations we make that point out interesting aspects and failure cases of our model.

---

**Input:** Arsenal , Newcastle United and Southampton have checked on Caen midfielder N'golo Kante . Paris-born Kante is a defensive minded player who has impressed for Caen this season and they are willing to sell for around £ 5million . Marseille have been in constant contact with Caen over signing the 24-year-old who has similarities with Lassana Diarra and Claude Makelele in terms of stature and style . N'Golo Kante is attracting interest from a host of Premier League clubs including Arsenal . Caen would be willing to sell Kante for around £ 5million .

---

**Reference:** n'golo kante is wanted by arsenal , newcastle and southampton . marseille are also keen on the £ 5m rated midfielder . kante has been compared to lassana diarra and claude makelele . click here for the latest premier league news .

---

**See et al. (2017) (+ Pointer):** arsenal , newcastle united and southampton have checked on caen midfielder n'golo kante . paris-born kante is attracting interest from a host of premier league clubs including arsenal . paris-born kante is attracting interest from a host of premier league clubs including arsenal

---

**See et al. (2017) (+ Pointer + Coverage):** arsenal , newcastle united and southampton have checked on caen midfielder n'golo kante . paris-born kante is a defensive minded player who has impressed for caen this season . marseille have been in constant contact with caen over signing the 24-year-old .

---

**BILSTM+GNN → LSTM:** marseille have been linked with caen midfielder %UNK% %UNK% . marseille have been interested from a host of premier league clubs including arsenal . caen have been interested from a host of premier league clubs including arsenal .

---

**BILSTM+GNN → LSTM+POINTER** n'golo kante is attracting interest from a host of premier league clubs . marseille have been in constant contact with caen over signing the 24-year-old . the 24-year-old has similarities with lassana diarra and claude makelele in terms of stature .

---

Figure 4: Sample natural language translations from the CNN-DM dataset.

**METHODDOC** Figures 1 and 3 illustrate typical results of baselines and our model on the METHODDOC task (see Appendix A for more examples). The hardness of the task stems from the large number of distractors and the need to identify the most relevant parts of the input. In Figure 1, the token "parameter" and variations appears many times, and identifying the correct relationship is non-trivial, but is evidently eased by graph edges explicitly denoting these relationships. Similarly, in Figure 3, many variables are passed around, and the semantics of the method require understanding how information flows between them.

**NLSUMMARIZATION** Figure 4 shows one sample summarization. More samples for this task can be found in Appendix B. First, we notice that the model produces natural-looking summaries with no noticeable negative impact on the fluency of the language over existing methods. Furthermore, the GNN-based model seems to capture the central named entity in the article and creates a summary centered around that entity. We hypothesize that the GNN component that links long-distance relationships helps capture and maintain a better "global" view of the article, allowing for better identification of central entities. Our model still suffers from repetition of information (see Appendix B), and so we believe that our model would also profit from advances such as taking coverage into account (See et al., 2017) or optimizing for ROUGE-L scores directly via reinforcement learning (Chen & Bansal, 2018; Narayan et al., 2018).

## 5 RELATED WORK

Natural language processing research has studied summarization for a long time. Most related is work on abstractive summarization, in which the core content of a given text (usually a news article) is summarized in a novel and concise sentence. Chopra et al. (2016) and Nallapati et al. (2016) use deep learning models with attention on the input text to guide a decoder that generates a summary. See et al. (2017) and McCann et al. (2018) extend this idea with pointer networks (Vinyals et al., 2015a) to allow for copying tokens from the input text to the output summary. These approaches treat text as a simple token sequences, not explicitly exposing additional structure. In principle, deep sequence networks are known to be able to learn the inherent structure of natural language (*e.g.* in parsing (Vinyals et al., 2015b) and entity recognition (Lample et al., 2016)), but our experiments indicate that explicitly exposing this structure by separating concerns improves performance.

Recent work in summarization has proposed improved training objectives for summarization, such as tracking coverage of the input document (See et al., 2017) or using reinforcement learning to directly identify actions in the decoder that improve target measures such as ROUGE-L (Chen & Bansal, 2018; Narayan et al., 2018). These objectives are orthogonal to the graph-augmented encoder discussed in this work, and we are interested in combining these efforts in future work.

Exposing more language structure explicitly has been studied over the last years, with a focus on tree-based models (Tai et al., 2015). Very recently, first uses of graphs in natural language processing have been explored. Marcheggiani & Titov (2017) use graph convolutional networks to encode single sentences and assist machine translation. De Cao et al. (2018) create a graph over named entities over a set of documents to assist question answering. Closer to our work is the work of Liu et al. (2018), who use abstract meaning representation (AMR), in which the source document is first parsed into AMR graphs, before a summary graph is created, which is finally rendered in natural language. In contrast to that work we do not use AMRs but directly encode relatively simple relationships directly on the tokenized text, and do not treat summarization as a graph rewrite problem. Combining our encoder with AMRs to use richer graph structures may be a promising future direction.

Finally, summarization in source code has also been studied in the forms of method naming, comment and documentation prediction. Method naming has been tackled with a series of models. For example, Allamanis et al. (2015) use a log-bilinear network to predict method names from features, and later extend this idea to use a convolutional attention network over the tokens of a method to predict the subtokens of names (Allamanis et al., 2016). Raychev et al. (2015) and Bichsel et al. (2016) use CRFs for a range of tasks on source code, including the inference of names for variables and methods. Recently, Alon et al. (2018b;a) extract and encode paths from the syntax tree of a program, setting the state of the art in accuracy on method naming.

Linking text to code can have useful applications, such as code search (Gu et al., 2018), traceability (Guo et al., 2017), and detection of redundant method comments (Louis et al., 2018). Most approaches on source code either treat it as natural language (*i.e.*, a token sequence), or use a language parser to explicitly expose its tree structure. For example, Barone & Sennrich (2017) use a simple sequence-to-sequence baseline, whereas Hu et al. (2017) summarize source code by linearizing the abstract syntax tree of the code and using a sequence-to-sequence model. Wan et al. (2018) instead directly operate on the tree structure using tree recurrent neural networks (Tai et al., 2015). The use of additional structure on related tasks on source code has been studied recently, for example in models that are conditioned on learned traversals of the syntax tree (Bielik et al., 2016) and in graph-based approaches (Allamanis et al., 2018; Cvitkovic et al., 2018). However, as noted by Liao et al. (2018), GNN-based approaches suffer from a tension between the ability to propagate information across large distances in a graph and the computational expense of the propagation function, which is linear in the number of graph edges per propagation step.

## 6 DISCUSSION & CONCLUSIONS

We presented a framework for extending sequence encoders with a graph component that can leverage rich additional structure. In an evaluation on three different summarization tasks, we have shown that this augmentation improves the performance of a range of different sequence models across all tasks. We are excited about this initial progress and look forward to deeper integration of mixed sequence-graph modeling in a wide range of tasks across both formal and natural languages. The key insight, which we believe to be widely applicable, is that inductive biases induced by explicit relationship modeling are a simple way to boost the practical performance of existing deep learning systems.

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

# A  CODE SUMMARIZATION SAMPLES

## A.1  METHODDOC

### C# Sample 1

```csharp
public static bool TryConvertTo(object valueToConvert, Type resultType,
                        IFormatProvider formatProvider, out object result){
    result = null;
    try{
        result = ConvertTo(valueToConvert, resultType, formatProvider);
    } catch (InvalidCastException){
        return false;
    } catch (ArgumentException){
        return false;
    }
    return true;
}
```

| | |
|---|---|
| **Ground truth** | sets result to valuetoconvert converted to resulttype considering formatprovider for custom conversions calling the parse method and calling convert . changetype . |
| **BILSTM → LSTM** | converts the specified type to a primitive type . |
| **BILSTM+GNN → LSTM** | sets result to resulttype |
| **BILSTM+GNN → LSTM+POINTER** | sets result to valuetoconvert converted to resulttype. |

### C# Sample 2

```csharp
public virtual Task Init(string name, IProviderRuntime providerRuntime,
                    IProviderConfiguration config){
    Log = providerRuntime.GetLogger(this.GetType().FullName);
    this.serializerSettings = OrleansJsonSerializer.GetDefaultSerializerSettings();
    return TaskDone.Done;
}
```

| | |
|---|---|
| **Ground truth** | initializes the storage provider |
| **BILSTM → LSTM** | creates a grain object |
| **BILSTM+GNN → LSTM** | initializes the provider provider |
| **BILSTM+GNN → LSTM+POINTER** | initialization function to initialize the specified provider. |

### C# Sample 3

```csharp
public void NullParameter(){
    TaskParameter t = new TaskParameter(null);
    Assert.Null(t.WrappedParameter);
    Assert.Equal( TaskParameterType.Null , t.ParameterType );
    ((INodePacketTranslatable) t).Translate(
                        TranslationHelpers.GetWriteTranslator());
    TaskParameter t2 = TaskParameter.FactoryForDeserialization(
                        TranslationHelpers.GetReadTranslator());
    Assert.Null(t2.WrappedParameter);
    Assert.Equal(TaskParameterType.Null, t2.ParameterType);
}
```

| | |
|---|---|
| **Ground truth** | verifies that construction and serialization with a null parameter is ok |
| **BILSTM → LSTM** | tests that the value is a value that is a value to the specified type |
| **BILSTM+GNN → LSTM** | verifies that construction with an parameter parameter |
| **BILSTM+GNN → LSTM+POINTER** | verifies that construction and serialization with a parameter that is null |

### C# Sample 4

```csharp
public override DbGeometryWellKnownValue CreateWellKnownValue(DbGeometry geometryValue){
    geometryValue.CheckNull("geometryValue");
    var spatialValue = geometryValue.AsSpatialValue();
    DbGeometryWellKnownValue result = CreateWellKnownValue(spatialValue,
```

```
      ()=>SpatialExceptions.CouldNotCreateWellKnownGeometryValueNoSrid("geometryValue"),
      ()=>SpatialExceptions.CouldNotCreateWellKnownGeometryValueNoWkbOrWkt("geometryValue"),
      (srid, wkb, wkt) => new DbGeometryWellKnownValue() {
        CoordinateSystemId = srid, WellKnownBinary = wkb, WellKnownText = wkt
      });
   return result;
}
```

| | |
|---|---|
| **Ground truth** | creates an instance of t:system.data.spatial.dbgeometry value using one or both of the standard well known spatial formats. |
| **BiLSTM** $\rightarrow$ **LSTM** | creates a t:system.data.spatial.dbgeography value based on the specified well known binary value . |
| **BiLSTM+GNN** $\rightarrow$ **LSTM** | creates a new t:system.data.spatial.dbgeography instance using the specified well known spatial formats . |
| **BiLSTM+GNN** $\rightarrow$ **LSTM+POINTER** | creates a new instance of the t:system.data.spatial.dbgeometry value based on the provided geometry value and returns the resulting well as known spatial formats . |

## A.2 METHODNAMING

### C# Sample 1

```
public bool _(D d) {
   return d != null && d.Val == Val ;
}
```

| | |
|---|---|
| **Ground truth** | equals |
| **BiLSTM** $\rightarrow$ **LSTM** | foo |
| **BiLSTM+GNN** $\rightarrow$ **LSTM** | equals |
| **BiLSTM+GNN** $\rightarrow$ **LSTM+POINTER** | equals |

### C# Sample 2

```
internal void _(string switchName, Hashtable bag, string parameterName) {
   object obj = bag[parameterName];
   if(obj != null){
      int value = (int) obj;
      AppendSwitchIfNotNull(switchName,
                       value.ToString(CultureInfo.InvariantCulture));
   }
}
```

| | |
|---|---|
| **Ground truth** | append switch with integer |
| **BiLSTM** $\rightarrow$ **LSTM** | set string |
| **BiLSTM+GNN** $\rightarrow$ **LSTM** | append switch |
| **BiLSTM+GNN** $\rightarrow$ **LSTM+POINTER** | append switch if not null |

### C# Sample 3

```
   internal static string _(){
      var currentPlatformString = string.Empty;
      if (RuntimeInformation.IsOSPlatform(OSPlatform.Windows)){
         currentPlatformString = "WINDOWS";
      }
      else if (RuntimeInformation.IsOSPlatform(OSPlatform.Linux)){
         currentPlatformString = "LINUX";
      }
      else if ( RuntimeInformation.IsOSPlatform(OSPlatform.OSX)) {
         currentPlatformString = "OSX";
      }
      else {
         Assert.True(false, "unrecognized current platform");
      }
      return currentPlatformString ;
}
```

| **Ground truth** | | get os platform as string |
|---|---|---|
| **BɪLSTM** | $\rightarrow$ **LSTM** | get name |
| **BɪLSTM+GNN** | $\rightarrow$ **LSTM** | get platform |
| **BɪLSTM+GNN** | $\rightarrow$ **LSTM+Pᴏɪɴᴛᴇʀ** | get current platform string |

### C# Sample 4

```csharp
public override DbGeometryWellKnownValue CreateWellKnownValue(DbGeometry geometryValue){
  geometryValue.CheckNull("geometryValue");
  var spatialValue = geometryValue.AsSpatialValue();
  DbGeometryWellKnownValue result = CreateWellKnownValue(spatialValue,
    ()=>SpatialExceptions.CouldNotCreateWellKnownGeometryValueNoSrid("geometryValue"),
    ()=>SpatialExceptions.CouldNotCreateWellKnownGeometryValueNoWkbOrWkt("geometryValue"),
    (srid, wkb, wkt) => new DbGeometryWellKnownValue () {
       CoordinateSystemId = srid , WellKnownBinary = wkb , WellKnownText = wkt
    });
  return result;
}
```

| **Ground truth** | | create well known value |
|---|---|---|
| **BɪLSTM** | $\rightarrow$ **LSTM** | spatial geometry from xml |
| **BɪLSTM+GNN** | $\rightarrow$ **LSTM** | geometry point |
| **BɪLSTM+GNN** | $\rightarrow$ **LSTM+Pᴏɪɴᴛᴇʀ** | get well known value |

### Java Sample 1

```java
public static void _(String name, int expected, MetricsRecordBuilder rb) {
   Assert.assertEquals("Bad value for metric " + name,
                 expected,
                 getIntCounter(name, rb));
}
```

| **Ground truth** | | assert counter |
|---|---|---|
| **BɪLSTM** | $\rightarrow$ **LSTM** | assert email value |
| **BɪLSTM+GNN** | $\rightarrow$ **LSTM** | assert header |
| **BɪLSTM+GNN** | $\rightarrow$ **LSTM+Pᴏɪɴᴛᴇʀ** | assert int counter |

## B    NATURAL LANGUAGE SUMMARIZATION SAMPLES

**Input:** -LRB- CNN -RRB- Gunshots were fired at rapper Lil Wayne 's tour bus early Sunday in Atlanta . No one was injured in the shooting , and no arrests have been made , Atlanta Police spokeswoman Elizabeth Espy said . Police are still looking for suspects . Officers were called to a parking lot in Atlanta 's Buckhead neighborhood , Espy said . They arrived at 3:25 a.m. and located two tour buses that had been shot multiple times . The drivers of the buses said the incident occurred on Interstate 285 near Interstate 75 , Espy said . Witnesses provided a limited description of the two vehicles suspected to be involved : a " Corvette style vehicle " and an SUV . Lil Wayne was in Atlanta for a performance at Compound nightclub Saturday night . CNN 's Carma Hassan contributed to this report .

**Reference:** rapper lil wayne not injured after shots fired at his tour bus on an atlanta interstate , police say . no one has been arrested in the shooting

**See et al. (2017) (+ Pointer):** police are still looking for suspects . the incident occurred on interstate 285 near interstate 75 , police say . witnesses provided a limited description of the two vehicles suspected to be involved : a " corvette style vehicle " and an suv .

**See et al. (2017) (+ Pointer + Coverage):** lil wayne 's tour bus was shot multiple times , police say . police are still looking for suspects . they arrived at 3:25 a.m. and located two tour buses that had been shot .

**BILSTM+GNN → LSTM:** the incident occurred on interstate %UNK% near interstate 75 . no one was injured in the shooting , and no arrests have been made , atlanta police spokeswoman says .

**BILSTM+GNN → LSTM+POINTER** gunshots fired at rapper lil wayne 's tour bus early sunday in atlanta , police say . no one was injured in the shooting , and no arrests have been made , police say .

---

**Input:** Tottenham have held further discussions with Marseille over a potential deal for midfielder Florian Thauvin . The 22-year-old has been left out of the squad for this weekend 's game with Metz as Marseille push for a £ 15m sale . The winger , who can also play behind the striker , was the subject of enquiries from Spurs earlier in the year and has also been watched by Chelsea and Valencia . Tottenham have held further talks with Ligue 1 side Marseille over a possible deal for Florian Thauvin . Marseille are already resigned to losing Andre Ayew and Andre-Pierre Gignac with English sides keen on both . Everton , Newcastle and Swansea , have all shown an interest in Ayew , who is a free agent in the summer .

**Reference:** florian thauvin has been left out of marseille 's squad with metz . marseille are pushing for a £ 15m sale and tottenham are interested . the winger has also been watched by chelsea and la liga side valencia .

**See et al. (2017) (+ Pointer):** tottenham have held further discussions with marseille over a potential deal for midfielder florian thauvin . the 22-year-old has been left out of the squad for this weekend 's game with metz as marseille push for a 15m sale .

**See et al. (2017) (+ Pointer + Coverage):** florian thauvin has been left out of the squad for this weekend 's game with metz as marseille push for a 15m sale . the 22-year-old was the subject of enquiries from spurs earlier in the year .

**BILSTM+GNN → LSTM:** the 22-year-old has been left out of the squad for this weekend 's game with metz . the 22-year-old has been left out of the squad for this weekend 's game with metz . the winger has been left out of the squad for this weekend 's game with metz .

**BILSTM+GNN → LSTM+POINTER** tottenham have held further discussions with marseille over a potential deal . the winger has been left out of the squad for this weekend 's game . tottenham have held further talks with marseille over a potential deal .

## C  CODE DATASETS INFORMATION

### C.1  C# DATASET

We extract the C# dataset from open-source projects on GitHub. Overall, our dataset contains 460,905 methods, 55,635 of which have a documentation comment. The dataset is split 85-5-10%. The projects and *exact* state of the repositories used is listed in Table 3

Table 3: Projects in our C# dataset. Ordered alphabetically.

| Name | Git SHA | Description |
|------|---------|-------------|
| Akka.NET | 6f32f6a7 | Actor-based Concurrent & Distributed Framework |
| AutoMapper | 19d6f7fc | Object-to-Object Mapping Library |
| BenchmarkDotNet | 57005f05 | Benchmarking Library |
| CommonMark.NET | f3d54530 | Markdown Parser |
| CoreCLR | cc5dcbe6 | .NET Core Runtime |
| CoreFx | ec1671fd | .NET Foundational Libraries |
| Dapper | 3c7cde28 | Object Mapper Library |
| EntityFramework | c4d9a269 | Object-Relational Mapper |
| Humanizer | 2b1c94c4 | String Manipulation and Formatting |
| Lean | 90ee6aae | Algorithmic Trading Engine |
| Mono | 9b9e4f4b | .NET Implementation |
| MsBuild | 7f95dc15 | Build Engine |
| Nancy | de458a9b | HTTP Service Framework |
| NLog | 49fdd08e | Logging Library |
| Opserver | 9e4d3a40 | Monitoring System |
| orleans | f89c5866 | Distributed Virtual Actor Model |
| Polly | f3d2973d | Resilience & Transient Fault Handling Library |
| Powershell | 9ac701db | Command-line Shell |
| ravendb | 6437de30 | Document Database |
| roslyn | 8ca0a542 | Compiler & Code Analysis & Compilation |
| ServiceStack | 17f081b9 | Real-time web library |
| SignalR | 9b05bcb0 | Push Notification Framework |
| Wox | 13e6c5ee | Application Launcher |

### C.2  JAVA METHOD NAMING DATASETS

We use the datasets and splits of Alon et al. (2018a) provided by their website. Upon scanning all methods in the dataset, the size of the corpora can be seen in Table 4. More information can be found at Alon et al. (2018a).

### C.3  PYTHON METHOD DOCUMENTATION DATASET

We use the dataset as split of Barone & Sennrich (2017) provided by their GitHub repository. Upon parsing the dataset, we get 106,065 training samples, 1,943 validation samples and 1,937 test samples. We note that 16.9% of the documentation samples in the validation set and 15.3% of the samples in test set have a sample with the identical natural language documentation on the training set. This

Table 4: The statistics of the extracted graphs from the Java method naming dataset of Alon et al. (2018a).

| Dataset | Train Size | Valid Size | Test Size |
|---------|-----------|-----------|-----------|
| Java – Small | 691,505 | 23,837 | 56,952 |

eludes to a potential issue, described by Lopes et al. (2017). See Allamanis (2018) for a lengthier discussion of this issue.

## C.4    GRAPH DATA STATISTICS

Below we present the data characteristics of the graphs we use across the datasets.

Table 5: Graph Statistics For Datasets.

| Dataset | Avg Num Nodes | Avg Num Edges |
|---|---|---|
| CNN/DM | 903.2 | 2532.9 |
| C# Method Names | 125.2 | 239.3 |
| C# Documentation | 133.5 | 265.9 |
| Java-Small Method Names | 144.4 | 251.6 |

