# OpenReview forum: "Structured Neural Summarization"
_ICLR.cc/2019/Conference_

### Official Review · AnonReviewer3 · 2018-10-31
**Limited novelty and missing some key experiments**

**Rating:** 7
**Confidence:** 3

**Review:**

Note: I changed my original score from 4 to 7 based on the new experiments that answer many of the questions I had about the relative performance of each part of the model. The review below is the original one I wrote before the paper changes.

# Positive aspects of this submission

- The intuition and motivation behind the proposed model are well explained.

- The empirical results on the MethodNaming and MethodDoc tasks are very promising.

# Criticism

- The novelty of the proposed model is limited since it is essentially adding an existing GGNN layer, introduced by Li et al. (2015), on top of an existing LSTM encoder. The most important novelty seems to be the custom graph representation for these sequence inputs to make them compatible with the GGNN, which should then deserve a more in-depth study (i.e. ablation study with different graph representations, etc).

- Since you compare your model performance against Alon et al. on Java-small, it should be fair to report the numbers on Java-med and Java-large as well.

- The "GNN -> LSTM+POINTER" experiment results are reported on the MethodDoc task, but not for MethodNaming. Reporting this number for MethodNaming is essential to show the claimed empirical superiority of the hybrid encoder compared to GNN only.

- I have doubts about the usefulness of the proposed model for natural language summarization, for the following reasons:

    - The comparison of the proposed model for NLSummarization against See et al. is a bit unfair, since it uses additional information through the CoreNLP named entity recognizer and coreference models. With the experiments listed in Table 1, there is no way to know whether the increased performance is due to the hybrid encoder design or due the additional named entity and coreference information. Adding the entity and coreference data in a simpler way (i.e. at the token embedding level with a basic sequence encoder) in the ablation study would very useful to answer that question.

    - In NLSummarization, connecting sentence nodes using a NEXT edge can be analogous to using a hierarchical encoder, as used by Nallapati et al. ("Abstractive Text Summarization using Sequence-to-sequence RNNs and Beyond", 2016). Ignoring the other edges of the GNN graph, what are the theoretical and empirical advantages of your method compared to this sentence-level hierarchical encoder?

    - Adding the coverage decoder introduced by See et al. to your model would have been very useful to prove that the current performance gap is indeed due to the simplistic decoder and not something else.

- How essential is the weighted averaging for graph-level document representation (Gilmer et al. 2017) compared to uniform averaging?

- A few minor comments about writing:
    - In Table 1, please put the highest numbers in bold to improve readability
    - On page 7, the word "summaries" is missing in "the model produces natural-looking with no noticeable negative impact"
    - On page 9, "cove content" should be "core content"

---

> ### Author Response · Authors · 2018-11-11
> **Response & Additional Experiments**
>
> Thanks for your detailed comments, which we will integrate your comments in the next version of our paper.
>
> On novelty:
>
> We agree that we are not contributing fundamentally new models here – indeed, we refrained from introducing a more complex architecture to make it easy to adopt this modeling approach. We believe that our work introduces a simple way to fuse state-of-the-art sequence (not only LSTMs, but /any/ sequence encoder) learning with reasoning enabled by domain-specific graph constructions. We have not found this idea in prior work, and our experiments show the value across three different tasks from different domains. We hope that other researchers can profit from our work by integrating similar techniques into their own architectures and believe that this deserves publication and wider dissemination.
>
> As discussed in our reply to all reviewers, we will run additional experiments on the CNN/DM to analyze the influence of different graph constructions.
>
>
> On GNN->LSTM+pointer on MethodNaming:
>
> We decided to show this ablation experiment only on the MethodDoc task for presentation reasons, but we will rerun the model and provide additional results on the MethodNaming task in our next revision.
>
>
> On comparison with Alon et al. 2018 on the Java-Large corpus:
>
> We did run these experiments but realized that we could obtain best results by models that “felt” like they had too much capacity. Further analysis of this behavior traced this to a problem with a duplication of samples in the dataset. For example, about 30.7% of files in the Java-Large are near-duplicates of other files in the corpus (across all folds), indicating that results on these datasets primarily measure overfitting to the data. We managed to train competitive models, but only by choosing very large sizes for the hidden dimensions (>1000) and removing dropout. In contrast, Java-Small only has 3.0% duplicates. We will clarify this in the next version of our paper. [This is similar to our experiences with the Barone & Sennrich dataset discussed in Sect. 4.1.2.]
>
>
> On NL Summarization and additional information:
>
> We agree that our model uses additional information that is not available to the pure sequence models – indeed, we believe that the ability to use this information is the core contribution of our work. Indeed, it is unclear how to add information from the CoreNLP parser to a standard sequence model (how, for example, are coreference connections represented?). As discussed in our reply to all reviews, we will run additional experiments to further elucidate this effect. Primarily, we will run an LSTM baseline that uses additional per-token information in the embedding of words, and additionally will introduce fresh tokens (“<REF1>”, …) to mark points at which references are made. If you had other comparisons in mind, please do react quickly, as these experiments do take a bit of time...
>
>
> On comparison with Nallapati et al. 2016:
>
> The structure of the “Next” tokens in the graph model resembles that of Nallapati et al. (2016). However, the core difference is in how message-passing GNNs work. In Nallapati et al. (2016) computing the representations this is truly hierarchical, I.e. information flows in one direction: sentence representations are computed, then these are combined into a document representation. In a GNN, messages are passed in both directions, and thus our per-sentence nodes also allow the exchange of information between different tokens in the same sentence. Hence, our model is more comparable to a hierarchical setting in which information can flow both up and down.
>
>
> On using coverage:
>
> We wanted to avoid the additional work for this experiment, since we believe that the improvements from adding a coverage mechanism are orthogonal to the ones provided by our model but will now run this and provide the results once the experiments have finished.
>
>
> On weighted averaging:
>
> In past experiments on a variety of datasets and tasks, we have found that weighted averaging helps compared to uniform averaging. We believe that this is due to the fact that weighted averaging acts as an attention-like mechanism that allows the model to pick the salient information from the graph while allowing the message-passing to “freely” transfer information. Since this is also the accepted method in the GNN literature (e.g. Gilmer et al. 2017) we did not further experiment with this design decision. As our compute resources are limited, we want to avoid rerunning this ablation on the CNN/DM dataset, but will provide additional experiments on the two smaller tasks.
>
>
> Please, let us know if these do not sufficiently address the concerns you raise in your review and what alternative experiments are missing.

---

> > ### Comment · AnonReviewer3 · 2018-11-14
> > **The rebuttal add some useful clarifications and proposed experiments**
> >
> > Thank you for your reply and useful clarifications. The additional experiments you proposed may greatly enhance the quality of your paper indeed. My rating is subject to change depending on the outcome of these experiments.

---

> > > ### Comment · Area_Chair1 · 2018-11-26
> > > **Please review experiments**
> > >
> > > The authors have posted new experimental results. Do you think that these have addressed some of your concerns?

---

> > > ### Author Response · Authors · 2018-11-26
> > > **Results of additional experiments**
> > >
> > > We have added the results of a number of additional experiments to more clearly evaluate the effect of our contribution. On natural language summarization, our new Table 2 shows that (a) additional semantic information seem to not help sequence-based models; (b) using only syntactical, but not semantical information in the Sequence GNN setting is helpful, but larger gains are made when semantic information is included; and (c) these results become even starker when considering a fairer baseline (using the same codebase as ours), instead of results from another paper (with its own specialized hyperparameter tuning).
> > >
> > > While we plan to provide these results eventually, we identified another issue in our implementation of the coverage mechanism (due to which loss was not correctly normalized), and so this may take some more days. However, we believe that while these additional results will further improve the experimental evaluation, they are not crucial to document the value of our contribution.
> > >
> > > Overall, we would kindly request that you reconsider your rating given the additional experimental results, or provide further guidance on how to improve the paper.

---

> > > > ### Comment · AnonReviewer3 · 2018-11-26
> > > > **Useful additions have been made to the experiments**
> > > >
> > > > In light of the extensive new experiments and their conclusions, I indeed think that this paper is now much stronger. I have changed my original score from 4 to 7.

---

### Official Review · AnonReviewer1 · 2018-11-02
**A straightforward improvement for abstractive summarization**

**Rating:** 6
**Confidence:** 4

**Review:**

STRUCTURED NEURAL SUMMARIZATION

Summary:

This work combines Graph Neural Networks with a sequential approach to abstractive summarization across both natural and programming language datasets. The extension of GNNs is simple, but effective across all datasets in comparison to external baselines for CNN/DailyMail, internal baselines for C#, and a combination of both for Java. The idea of applying a more structured approach to summarization is well motivated given that current summarization methods tend to lack the consistency that a structured approach can provide. The chosen examples (which I hope are randomly sampled; are they?) do seem to suggest the efficacy of this approach with that intuition.

Comments:

Should probably cite CNN/DailyMail when it is first introduced as NLSummarization in Section 2 like you do the other datasets.

Can you further elaborate on how your approach is similar to and differs from that in Marcheggiani et al 2017 on Graph CNNs for Semantic Role Labeling, Bastings et al 2017 on Graph Convolutional Encoders for Syntax-aware Machine Translation, and De Cao et al 2018? Why should one elect to go the direction of sequential GNNs over the GCNs of those other works, and how might you compare against them? I would like to see some kind of ablation analysis or direct comparison with similar methods if possible.

Why would GNNs hurt SelfAtt performance on MethodDoc C# SelfAtt+GNN / SelfAtt?

Why not add the coverage mechanism from See et al 2017 in order to demonstrate that the method does in fact surpass that prior work? I'm left wondering whether the proposed method's returns diminish once coverage is added.

---

> ### Author Response · Authors · 2018-11-11
> **Response & Additional Experiments**
>
> Thanks for your thoughtful review and your time. As discussed in our reply to all reviews, we will run four additional experiments covering points raised by the different reviewers.
>
>
> On related work in NLP with graphs:
>
> Thank you for bringing up additional related work. The cited works handle quite different tasks, and so drawing a direct comparison to our work is hard. Marcheggiani et al. (2017) uses their model, with a single GCN propagation, for classification not sequence prediction, whereas Bastings et al. (2017) does sentence-to-sentence translation. Both employ purely syntactic graphs and thus lack the advantages that additional semantic information can provide. Our additional experiments 2 and 3 are designed to show the effect of this. The short paper of De Cao et al. (2018) uses a GCN over entities in multiple documents.  Finally, we want to highlight that we propose to use graphs for longer documents, whereas the approaches above are primarily concerned with single sentences. On average the CNN/DM documents lead to graphs with 900 nodes and 2.5k edges.
>
> Regarding the question of SequentialGNN vs GCN, we believe that there are no substantial differences between the use of GCNs and GGNNs. The core contribution proposed in our paper is the idea to fuse information obtained from state-of-the-art sequence models with a form of structured reasoning that can integrate domain knowledge.
> We will clarify the above in the related work section.
>
>
> On the performance of SelfAtt vs. SelfAtt+GNN on MethodDoc C#:
>
> In the paper, we discuss this result explicitly in the third paragraph of 4.1.4. The core reason for the decrease in ROUGE scores is that the SelfAtt+GNN model produces substantially longer outputs, which tends to impact ROUGE scores. This causes the substantial improvement in the BLEU score. We will extend the appendix to include examples of outputs of the SelfAtt/SelfAtt+GNN models that illustrate how the longer output improves the information content of the results. Overall, we want to note that ROUGE and BLEU are problematic measures for these tasks, but we are not aware of any other metrics that can be computed at scale.
>
>
> On randomness of shown samples:
>
> The sample in Figure 2 is one appearing in See et al. For Figure 1, we had to pick a sample that would fit within the given space, so it’s not randomly sampled. All other examples are randomly selected.

---

> > ### Comment · Area_Chair1 · 2018-11-19
> > **Follow up question regarding related work**
> >
> > Hello!
> > I had a follow-up question regarding related work: even given the response it still wasn't clear to me the differences and advantages of the proposed method, both theoretically and empirically, compared to previous work incorporating graph structures on the input side of sequence-to-sequence models. Even if the task is different, the methodology seems like it would be largely similar, so these methods would be reasonable baselines. Without a comparison it makes it a bit difficult to tell the merit of this particular work. Would you mind elaborating?

---

> > > ### Author Response · Authors · 2018-11-20
> > > **Answer regarding related work**
> > >
> > > The overall concept in Marcheggiani and Titov's work is similar, but we generalise it in four ways:
> > >  (1) We consider a wider range of sequence encoders.
> > >  (2) We show that the resulting GNN structure is useful for sequence decoding, with attention over the generated inputs.
> > >  (3) We consider a wider range of different tasks, with different graph structures.
> > >  (4) We incorporate semantic and across-sentence relationships, instead of only syntactic relationships.
> > >
> > > While this work tackles the same problem as we do (namely, modeling long-distance
> > > relationships in NLP) and uses the same fundamental idea (namely, modeling
> > > relationships in graphs), we feel that our work provides the empirical evidence
> > > that the idea is widely applicable, both across diverse modelling choices and
> > > task choices.
> > >
> > > Bastings et al. provide a follow-up on that work, focusing on aspect (2), adding
> > > a sequence decoder. Similarly, De Cao et al. build on a similar idea, but focus
> > > on aspect (4), but do not introduce intra-document relationships, but instead use
> > > the graph structure to reflect an entity graph. This does not use end-to-end training
> > > for the sequential structure of the natural language (they use pre-trained, fixed
> > > ELMo).
> > >
> > >
> > > Overall, we believe our contribution to generalise in all dimensions (1)-(4), hopefully
> > > providing enough experimental evidence so that all researchers working on sequential
> > > data with some inherent structure will consider mixed sequence/graph models in the
> > > future. This is why we included non-natural language tasks (but with obvious graph
> > > structure), showing the wide applicability of the idea.

---

### Official Review · AnonReviewer2 · 2018-11-06
**Interesting idea and promising results**

**Rating:** 6
**Confidence:** 4

**Review:**

This paper presents a structural summarization model with a graph-based encoder extended from RNN. Experiments are conducted on three tasks, including generating names for methods, generating descriptions for a function, and generating text summaries for news articles. Experimental results show that the proposed usage of GNN can improve performance by the models without GNN. I think the method is reasonable and results are promising, but I'd like to see more focused evaluation on the semantics captured by the proposed model (compared to the models without GNN).

Here are some questions and suggestions:

- Overall, I think additional evaluation should be done to evaluate on the semantic understanding aspects of the methods. Concretely, the Graph-based encoder has access to semantic information, such as entities. In order to better understand how this helps with the overall improvement, the authors should consider automatic evaluation and human evaluation to measure its contribution. Also from fig. 3, we can see that all methods get the "utf8 string" part right, but it's hard to say the proposed method generates better description.

- In the last table in Tab. 1, why the authors don't have results for adding GNN for the pointer-generator model with coverage?

---

> ### Author Response · Authors · 2018-11-11
> **Response & Additional Experiments**
>
> Thanks for your time and helpful comments. As discussed in our reply to all reviews, we will run four additional experiments covering points raised by the different reviewers. However, while we believe that a human evaluation of generated summaries would be helpful, setting this up during the rebuttal period seems to be impossible. Do let us know if you want us to run more experiments / provide more results.

---

### Author Response · Authors · 2018-11-11
**Response Summary**

Thank you for all your comments we respond to your comments individually. Below you can find a summary for all the reviewers.

We plan to run the following experiments:
•	[Experiment 1] BiLSTM on natural language inputs using Stanford CoreNLP information. For this, we will extend token embeddings by information from the CoreNLP parser, and introduce special tokens (“<REF1>”, …) to mark co-references.
•	[Experiment 2] BiLSTM+GNN on natural language inputs using only syntactic information. Concretely, each token will be represented by one node and we introduce one node per sentence. The only edges will be “NextToken” and “NextSentence”. This experiment tests the performance of our model using only syntactic information used by other models (e.g., hierarchical representations that split sentences).
•	[Experiment 3] BiLSTM+GNN on natural language input using syntactic and equality information. This is like experiment 2, but will also add edges between non-stopword nodes corresponding to tokens that have identical string representations when stemmed.
•	[Experiment 4] BilSTM+GNN -> LSTM+Pointer+Coverage. We will extend the full model by See et al. with additional graph information.

Please, do let us know if these sufficiently address the concerns you mention in your review or if you would like to see other experiments.

We also want to emphasize again the broad applicability of our method. While the natural language summarization task is clearly the most interesting one, we do want to remark that our very general model is able to compete with (and beat) specialized approaches on the source code tasks. We have spent very little optimizing our models to the different tasks, and strongly believe that intensive tuning of hyperparameters to each of these tasks could further improve our results.

---

### Author Response · Authors · 2018-11-23
**Updates to Paper and Results of Experiments**

We have updated the paper with additional experiments that the reviewers showed interest in (Table 2). Unfortunately, due to a problem with integrating the “coverage” idea of See et al. into our codebase and the long training times for these models, we were unable to update the paper with these results so far. However, we expect to provide these results over the next few days. We note that the time since the start of the rebuttal period was too short to do hyperparameter optimization on the CNN/DailyMail summarization dataset, and we instead used the same hyperparameters we used before (without coverage).

Concretely, we provide the following additional experimental results:

•	We have added the GNN->LSTM+pointer model for the Method Naming task as requested by Reviewer 3. The results show that removing the biLSTM encoder worsens the results compared to our biLSTM+GNN->LSTM+pointer network.

•	We ran biLSTM encoder-based baselines for the CNN/DM task using the OpenNMT implementation, to better compare with our extension of that codebase. Despite using the same setup as See et al. our experiments yield slightly worse results. This is most likely due to the fact that we have not performed separate hyperparameter optimization for each task but instead use identical hyperparameters for _all_ our tasks and datasets. Our biLSTM+GNN results are most fairly compared to these baselines.

•	As discussed in our last post, we have performed experiments 1-3 on CNN/DM to analyze the influence of the extra information provided by the CoreNLP parser. The results can be summarized as follows:

      o	[Experiment 1] We ran a biLSTM encoder with access to the CoreNLP parse information. Concretely, we extended the token embedding with an embedding of the per-token information provided by the parser, and additionally added tags marking references using fresh “<REF1>”, “<REF2>”, … tokens. Our results indicate that this only minimally improves results compared to the standard biLSTM encoder operating on words, and hence that exposing the structure explicitly by using a GNN encoder provides more advantages.

      o	[Experiment 2] Removing all linguistic structure, i.e. Stanford CoreNLP edges, but retaining the extra sentence nodes and edges, yields a small improvement over the baseline biLSTM-based encoder, increasing ROUGE-2 score by one point and yielding minor differences in the other metrics.

      o	[Experiment 3] When adding edges that connect tokens with identical stemmed string representations, the performance increases a bit but does not reach the performance levels comparable to using the full coreference resolution.


We have clarified the above points in the text. In conclusion, pending on the coverage experiment, the above experiments demonstrate that:

a)	Neither biLSTM, nor GNN encoders alone achieve best performance in summarization tasks and the biLSTM+GNN combination improves on all baselines in all cases.

b)	Encoding additional linguistic structure is helpful for natural language summarization but cannot be captured adequately using only a standard biLSTM encoder.

c)	Encoding sentence and connecting long-distance tokens with the same stems only slightly helps performance, while using more advanced resolution of references yields bigger gains.

Finally, we would like to emphasize again the broad applicability of our summarization method to both natural language and source code. While the natural language summarization task is clearly the most interesting for the reviewers, our summarization model is also able to compete with (and beat) specialized approaches on the two source code tasks on three datasets.

---

### Meta-Review · Area_Chair1 · 2018-12-13
**Minor novelty, extensive and informative empirical comparison**

**Confidence:** 2
**Recommendation:** Accept (Poster)

**Metareview:**

This paper examines ways of encoding structured input such as source code or parsed natural language into representations that are conducive for summarization. Specifically, the innovation is to not use only a sequence model, nor only a tree model, but both. Empirical evaluation is extensive, and it is exhaustively demonstrated that combining both models provides the best results.

The major perceived issue of the paper is the lack of methodological novelty, which the authors acknowledge. In addition, there are other existing graph-based architectures that have not been compared to.

However, given that the experimental results are informative and convincing, I think that the paper is a reasonable candidate to be accepted to the conference.